# Adaptive Multiple Optimal Learning Factors for Neural Network Training

## Abstract

This paper presents the Adapt-MOLF algorithm that merges the strengths of second order algorithms while addressing their limitations. Adapt-MOLF algorithm dynamically adjusts the number of weight groups per hidden unit to maximize error change per multiplication, optimizing computational efficiency. Leveraging curvature-based grouping and Gauss-Newton updates, it efficiently interpolates the Hessian and negative gradients for computation. The two-stage algorithm alternately determines output weights and employs multiple learning factors to train input weights in a Multi-Layer Perceptron. This adaptive adjustment of learning factors maximizes the error decrease per multiplication, showcasing superior performance over OWO-MOLF and Levenberg Marquardt (LM) across diverse datasets. Extensive experiments demonstrate its competitive or superior results compared to state-of-the-art algorithms particularly excelling in reducing testing errors. This research represents a promising advancement in second-order optimization methods for neural network training, offering scalability, efficiency, and superior performance across heterogeneous datasets.

## 1 Introduction

A Neural Network is a computational cornerstone crucial for function approximation and pattern recognition tasks. Its significance stems from inherent capabilities enabling universal approximation (1) and the approximation of Bayes discriminants (1) (2). Within the realm of neural network optimization, techniques like Adam (3), AdaDelta (4), AdaGrad (5), RmsProp (6), and Nesterov momentum (7) predominantly align with first-order training algorithms. However, a conspicuous gap persists in the literature concerning advanced optimization strategies for second-order training algorithms. The crux of the challenge with second-order algorithms lies in their scalability issues, predominantly rooted in the resource-intensive computation of Hessian matrices. As network size expands, the computational demands for Hessian matrix operations grow exponentially, presenting substantial scalability hurdles.

Neural networks have seen a re-birth after their increased success in the recent past. Their capacity for universal approximation (1) challenges the No-Free-Lunch theorem (8), implying that multi-layer perceptrons (MLPs) can effectively approximate diverse discriminants. While Adafactor (9) combines Adam optimization and memory efficiency to reduce memory overhead, its adaptability across various neural network architectures and datasets remains underexplored. AdaSmooth (10) introduces adaptable window sizes for past gradient accumulation but lacks investigation into its efficacy across dynamic or noisy datasets. Similarly, LAMB (11) showcases layer-wise weight updating for faster model training, enabling faster training of models like BERT (12), yet its scalability across diverse architectures and datasets demands deeper examination. Techniques like SLAMB (13), integrating LAMB with sparse GPU communication, show promise in accelerating large batch training but require further exploration regarding practical implementation challenges and hardware requirements. Additionally, the Symbolic Discovery of Optimization Algorithms (LION) (14) offers a comprehensive exploration of optimization algorithms but lacks comparative analyses against existing strategies, limiting insights into their relative effectiveness.

Shifting focus to second-order optimization techniques, Shampoo (15) introduces efficient gradient preconditioning using matrices instead of Hessian computations. However, Shampoo's scalability concerns and its

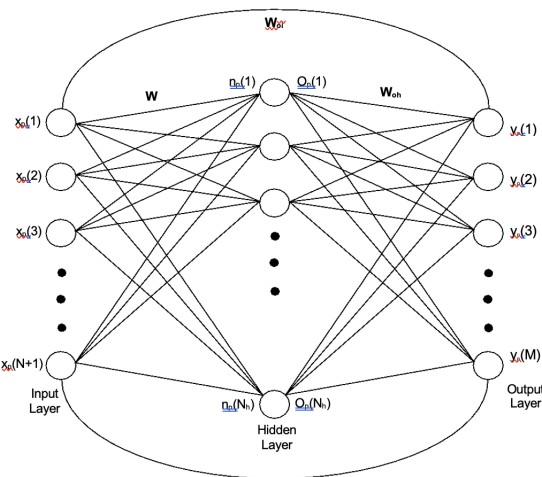

Figure 1: Fully Connected Multilayer Perceptron

performance in scenarios with highly non-convex functions or noisy gradients remain areas that necessitate further investigation to ascertain its generalizability. Moreover, Scalable Second Order Optimization in (16) efficiently implements the Shampoo algorithm. Yet, a comprehensive examination of its computational complexity and comparison with other scalable second-order techniques is lacking, hindering a complete understanding of its advantages over alternative methodologies. AdaHessian (17) innovatively utilizes only the diagonal elements of the Hessian matrix, reducing memory requirements significantly. However, the trade-offs between computational efficiency and optimization accuracy in complex, highly non-linear optimization landscapes warrant deeper analysis to comprehend its viability across diverse neural network architectures and training scenarios. The algorithm proposed in this paper addresses this challenge by optimizing neural network training operations more efficiently. The methodology involves segmenting the input weights of a multilayer perceptron (MLP) into clusters, followed by the application of Newton's method to compute a vector of learning factors, one for each cluster.

## 2 Background

We start by describing the multi layer perceptron (MLP), which is a non linear signal processor that has good approximation and classification properties. The MLP has basis functions that can adapt during the training process by utilizing example input and desired outputs.

### 2.1 Structure and Notation

The architecture of a fully connected feed forward multi layer perceptron (MLP) is shown in Figure 1. The input weights $w(k, n)$ connect the $n^{th}$ input to the $k^{th}$ hidden unit. Output weights $w_{oh}(m, k)$ connect the $k^{th}$ hidden unit's non-linear activation $O_p(k)$ to the $m^{th}$ output $y_p(m)$, which has a linear activation. The bypass weights $w_{oi}(m, n)$ connects the $n^{th}$ input to the $m^{th}$ output. The training data, described by the set of independent, identically distributed input-output pair $\{\mathbf{x_p}, \mathbf{t_p}\}$ consists of $N$ dimensional input vectors $\mathbf{x_p}$ and M dimensional desired output vectors, $\mathbf{t_p}$. The pattern number $p$ varies from 1 to $N_v$, where $N_v$ denotes the number of training vectors present in the datasets. Let $N_h$ denote the number of hidden units. Input bias is added by augmenting the input units with an extra element $x_p(N + 1)$, where $x_p(N + 1) = 1$. For each training pattern $p$, the hidden layer net function vector $\mathbf{n_p}$ can be written as $\mathbf{n_p} = \mathbf{W} \cdot \mathbf{x_p}$. The $k^{th}$ element of the hidden unit activation vector $\mathbf{O_p}$ is calculated as $O_p(k) = f(n_p(k))$ where $f(\cdot)$ denotes the sigmoid activation function. The network output vector, $\mathbf{y_p}$ can be written as

$$\mathbf{y_p} = \mathbf{W_{oi}} \cdot \mathbf{x_p} + \mathbf{W_{oh}} \cdot \mathbf{O_p} \tag{1}$$

The expression for the actual outputs given in equation (1) can be re-written as $\mathbf{y_p} = \mathbf{W_o} \cdot \mathbf{X_a}$, where $\mathbf{X_a} = [\mathbf{x_p}^T : \mathbf{O_p}^T]^T$ is the augmented input column vector with $N_u$ basis functions, where $N_u = 1 + N + N_h$. Similarly, $\mathbf{W_o}$ is the $M$ by $N_u$ dimensional augmented weight matrix defined as $\mathbf{W_o} = [\mathbf{W_{oh}}, : \mathbf{W_{oi}}]$. The training process for a MLP involves minimizing the mean squared error (3) between the desired and actual network output. This optimization requires fine-tuning the network's weights and biases to ensure a close match between computed and desired outputs. To train an MLP effectively, we reformulate the learning problem as an optimization task within a structural risk minimization framework (18; 19). This framework aims to minimize the error function E, represented by equation 3, acting as a surrogate for non-smooth classification errors. From a Bayesian perspective (18), the approach involves maximizing the likelihood function or minimizing the mean square error (MSE) in a least square sense. The MSE, calculated between inputs and outputs, is defined as:

$$E = \frac{1}{N_v} \sum_{p=1}^{N_v} \sum_{i=1}^{M} [t_p(i) - y_p(i)]^2 \tag{3}$$

The nonlinearity within $\mathbf{y_p}$ introduces non-convexity into the error function E, potentially leading to local minima in practice. We assume $\mathbf{t_p}$ follows a Gaussian distribution concerning input $\mathbf{x_p}$. Our aim is to determine the optimal weights within the MLP structure. Employing the empirical risk minimization framework (20), we design learning algorithms, benefiting from the advantage of transforming MLP training into an optimization problem. This conversion allows us to leverage various optimization algorithms to enhance MLP learning processes.

## 2.2 MLP Initialization

As from (20), the input weights matrix $\mathbf{W}$ is initialized randomly from a zero mean Gaussian random number generator. The training of the input weights, strongly depends on the gradient of the hidden units activation functions with respect to the inputs. Training of input weights will cease if the hidden units it feeds into has an activation function derivative of zero for all patterns. In order to remove the dominance of a large variance inputs, we divide the input weights by the input's standard deviation. Therefore we adjust the mean and standard deviation of all the hidden units net functions. This is called net control as in (21). At this point, we have determined the initial input weights and we are now ready to initialize the output weights. To initialize the output weight matrix $\mathbf{W_o}$, we use output weight optimization (OWO) (20). OWO minimizes the error function from equation (3) with respect to $\mathbf{W_o}$ by solving the $M$ sets of $N_u$ equations in $N_u$ unknowns given by

$$\mathbf{C} = \mathbf{R} \cdot \mathbf{W_o^T} \tag{2}$$

where cross-correlation matrix $\mathbf{C} = \frac{1}{N_v} \sum_{p=1}^{N_v} \mathbf{t_p} \cdot \mathbf{X_a^T}$, auto-correlation matrix $\mathbf{R} = \frac{1}{N_v} \sum_{p=1}^{N_v} \mathbf{X_a} \cdot \mathbf{X_a^T}$. In terms of optimization theory, solving equation (2) is merely Newton's algorithm for the output weights (22). After initialization of $\mathbf{W}$, $\mathbf{W_{oi}}$, $\mathbf{W_{oh}}$, we begin a two step procedure in which we modify $\mathbf{W}$ and perform OWO to modify $\mathbf{W_o}$. The MLP network is now initialized and ready to be trained with first or second order algorithms. Training an MLP can be seen as an unconstrained optimization problem that usually involves first order gradient methods such as backpropagation (BP), conjugate gradient (CG) and second order Levenberg-Marquardt (LM), Newton's method as the most popular learning algorithm. Training algorithms can be classified as One Stage, in which all the weights of the network are updated simultaneously and Two Stage, in which input and output weights are trained alternately.

## 2.3 Optimization algorithms

First-order optimization algorithms focus solely on the utilization of the first derivative to iteratively improve convergence. Contrasting these, the concept of employing a second-order method involves enhancing first-order algorithms by incorporating both the first and second derivatives (20). In our study, we'll compare our work with scaled conjugate gradient, an optimization algorithms positioned between first and second-order optimization techniques. Alongside these, we briefly review two, second order algorithms namely, single-stage

Levenberg-Marquardt (LM) (23) and a two-stage Output Weight Optimization-Multiple Optimal Learning Factor (OWO-MOLF).

### 2.3.1 Scaled conjugate Gradient

In a steepest descent algorithm, the weights are updated in the negative gradient direction. Although the error function reduces most rapidly along the negative direction of the gradient, it does not necessarily create fast convergence. Conjugate gradient algorithm (24) performs a line-search in the conjugate direction and has faster convergence than backpropagation algorithm. Although scaled conjugate gradient (SCG) is a general unconstrained optimization technique, its use in efficiently training an MLP is well documented in (25). To train an MLP using conjugate gradient algorithm, we use a direction vector that is obtained from the gradient $\mathbf{g}$ as $\mathbf{p} \leftarrow -\mathbf{g} + B_1 \cdot \mathbf{p}$. Here $\mathbf{p} = \text{vec}(\mathbf{P}, \mathbf{P_{oh}}, \mathbf{P_{oi}})$ and $\mathbf{P}$, $\mathbf{P_{oi}}$ and $\mathbf{P_{oh}}$ are the direction vectors. $B_1$ is the ratio of the gradient energy from two consecutive iterations. This direction vector, in turn, update all the weights simultaneously as $\mathbf{w} \leftarrow \mathbf{w} + z \cdot \mathbf{p}$. The Conjugate Gradient algorithm possesses several advantages. Firstly, its convergence rate aligns with the number of unknowns. Secondly, it outperforms the steepest descent method and can handle nonquadratic error functions. Thirdly, as it avoids matrix inversion involving Hessians, its computational cost remains at O($\mathbf{w}$), where $\mathbf{w}$ represents the weight vector's size. For a detailed pseudocode, refer to (20). In neural network training, the learning factor z determines the convergence speed. Typically, a small positive z works but leads to slow convergence, while a large z might increase the error E (20). To counter this, heuristic scaling methods adjust learning factors between iterations to hasten convergence. However, an Optimal Learning Factor (OLF) for OWO-BP (Output Weight Optimization Backpropagation) can be derived non-heuristically using Taylor's series for error E as discussed in (20).

### 2.3.2 Levenberg-Marquardt algorithm

The LM algorithm is a compromise between Newton's method, which converges rapidly near local or global minima but may diverge, and gradient descent, which has assured convergence through a proper selection of step size parameter but converge slowly. The LM algorithm is a sub-optimal method as usually $\mathbf{H}$ is singular in Newton's method, an alternate is to modify the Hessian matrix as in LM (26) algorithm or use a two step method such as layer by layer training (27). In LM, we modify the Hessian as $\mathbf{H_{LM}} = \mathbf{H} + \lambda \cdot \mathbf{I}$. Here $\mathbf{I}$ is the identity matrix of the same dimensions as $\mathbf{H}$ and $\lambda$ is a regularizing parameter that forces the sum matrix ($\mathbf{H} + \lambda \cdot \mathbf{I}$) to be positive definite and safely well conditioned throughout the computation. We calculate the second order direction, $\mathbf{d}$, similar to Newton's method as $\mathbf{H_{LM}} \cdot \mathbf{d} = \mathbf{g}$. Upon acquiring $\mathbf{H_{LM}}$, the model's weights undergo an update. The regularization parameter $\lambda$ significantly influences the LM algorithm's behavior. For optimal $\lambda$ selection, (28) propose an excellent *Marquardt recipe*. However, practically, computing $\mathbf{H_{LM}}$ can be computationally demanding, especially in high-dimensional weight vectors $\mathbf{w}$. Consequently, due to scalability limitations, LM proves more suitable for smaller networks. A detailed exploration of the LM algorithm can be found in (20).

### 2.3.3 OWO-MOLF

An alternative to LM, the OWO-MOLF algorithm adopts a two-stage "layer by layer" approach, leveraging Output Weight Optimization (OWO) and Multiple Optimal Learning Factors (MOLF) per hidden unit (29). Unlike heuristic methods relying on various learning rates or momentum terms (30; 31), OWO-MOLF targets enhanced learning speed and convergence.This algorithm handles input weight updates via the negative gradient matrix G and a size ($N_h$,1) vector of learning factors z. Meanwhile, output weights are trained using OWO. A key concept involves updating input weight matrices for each epoch, replacing a single optimal learning factor $z$ with a vector $\mathbf{z}$ estimated using Newton's method (23). This vector solves $\mathbf{H_{molf}} \cdot \mathbf{z} = \mathbf{g_{molf}}$ through orthogonal least squares (OLS), where $\mathbf{H_{molf}}$ and $\mathbf{g_{molf}}$ represent the Hessian and negative gradient, respectively, concerning the error with respect to $\mathbf{z}$. Further implementation details are available in (20).

### 2.4 Challenges and Objectives

The paper addresses the following challenges posed by second order algorithms. Algorithms such as Lavenberg-Marquardt incur significant computational costs due to their reliance on optimizing all weights utilizing second

order information. In contrast, algorithms like OWO-MOLF, Shampoo, and Ada-Hessian attempt to mitigate these computational demands by either optimizing a subset of parameters via second order methods or by employing a lower rank approximation of the Hessian matrix. This approach reduces the need to compute and invert the full Hessian matrix for all parameters. However, a common limitation within these algorithms, when applied to a specific model architecture, is their consistent optimization of a similar fraction of parameters using second order methods for every training iteration for a given data-set. This uniform strategy leads to an unnecessary escalation in computational requirements. The objective of this paper is to formulate a novel second order training algorithm that dynamically adjusts the number of parameters optimized using second order information. This adjustment will be specific to each training iteration, dependent on the dataset and model architecture involved. The goal is to enhance computational efficiency without increase in optimization time. In order to address the objectives, we introduce Adapt-MOLF, a second-order algorithm that involves creating groups for each hidden unit, computing a single learning factor for each group.

## 3 Adapt-MOLF

The OWO-Newton algorithm often has excellent error convergence characteristics, but its performance decreases when there are linear dependencies in the input signal or hidden unit activations. Additionally, when the error function deviates from a quadratic form, the algorithm's performance is compromised. Moreover, OWO-Newton is computationally more expensive than first order algorithms. In contrast, the OWO-MOLF algorithm, while not matching the convergence speed per iteration of OWO-Newton, exhibits robustness against linearly dependent input signals and operates with lower computational overhead (22). The main objective of our paper is to integrate the strengths of OWO-MOLF and OWO-Newton while addressing their limitations. Adapt-MOLF dynamically adapts between OWO-MOLF and OWO-Newton, aiming to retain their respective advantages. In OWO-MOLF, $N_h$ learning factors correspond to each hidden unit, whereas in the Adapt-MOLF algorithm, the number of computed learning factors ranges from $N_h$ to $N_h \cdot (N + 1)$ in each iteration. Adapt-MOLF dynamically tailors second-order information use, optimizing computational efficiency across various scenarios.

### 3.1 Adaptive grouping of the input weights

In Adapt-MOLF, a *group* refers to a collection of input weights updated using a shared learning factor. For instance, in the steepest descent method, a single learning factor $z$ updates all network weights within one group (20). In OWO-MOLF, the input weights linked to a hidden unit are updated using a learning factor specific to that unit, forming groups of size $N + 1$, totaling one group per hidden unit ($N_g = 1$) (22). However, in Adapt-MOLF, group sizes fluctuate between $N + 1$ and 1, varying the number of groups per hidden unit from 1 to $N + 1$. Here, input weights $\mathbf{W}$ are grouped based on the curvature $\mathbf{L}$ of the error function concerning these weights. The elements of the matrix $L$ are

$$l(k, n) = \frac{\partial^2 E}{\partial w(k, n)^2} = \frac{2}{N_v} \sum_{i=1}^{M} w_{oh}(i, k)^2 \sum_{p=1}^{N_v} f'(n_p(k))^2 x_p(n)^2 \tag{3}$$

### 3.2 Adaptive grouping for error-optimized multiplication

Another adaptive approach we study is to optimize the number of groups per hidden unit to maximize error change per multiplication. In each iteration, we compute the error change by evaluating the difference between the current and previous iteration errors. This calculated error change is then divided by the number of multiplications required in the current iteration, defining the *error per multiply* (EPM) for that specific iteration using equation (4):

$$EPM(i_t) = \frac{E(i_t - 1) - E(i_t)}{M(i_t)} \tag{4}$$

Here, $M(i_t)$ denotes the number of multiplications in iteration $i_t$, while $EPM(i_t)$ represents the error per multiply for that particular iteration. As the error change per multiply fluctuates, the number of groups per hidden unit ($N_g$) dynamically adjusts. An increase in error change per multiply leads to an increase in $N_g$, and conversely, a decrease in error change per multiply results in a reduction of $N_g$. This key aspect of Adapt-MOLD gives substantial error reduction during initial iterations, operating akin to the OWO-Newton algorithm. As convergence to local minima occurs, the algorithm dynamically shifts towards behavior resembling the OWO-MOLF algorithm, reducing computational complexity while fine-tuning convergence.

### 3.3 Adaptive learning factor using compact hessians

Consider an MLP where the input weights are trained using negative gradients. Rather than a single learning factor, envision a vector of learning factors represented by $z$, with elements $z_{k,c}$ employed to update all weights $w(k,n)$ pertaining to group $c$ of hidden unit $k$. The error function to minimize from 3. The predicted output $y_p(i)$ is given by

$$y_p(i) = \sum_{n=1}^{N+1} x_p(n)w_{oi}(n) + \sum_{k=1}^{N_h} w_{oh}(i,k)f\left(\sum_{c=1}^{N_g}\sum_{a\in c} x_p(i_k(a))\left[(w(k,i_k(a))) + z_{k,c}g(k,i_k(a))\right]\right) \tag{41}$$

where, $N_g$ is the number of groups per hidden unit, $c$ is the group index. $I_k$ is the vector of input indices of weights connected to hidden unit $k$ sorted in descending order of curvature computed using equation (3). $I_k = [n_1, n_2, n_3 \ldots n_{N+1}]$ where $n_1, n_2, n_3 \ldots n_{N+1}$ are input indices such that $l(k,n_1) \geq l(k,n_2) \geq l(k,n_3) \ldots \geq l(k,n_{N+1})$. $z_{k,c}$ is the learning factor used to update all the input weights that belong to group c of hidden unit k. The total number of learning factors that are going to be computed is $L = N_h \cdot N_g$. Where, $L$ can vary from $N_h$ to $N_h \cdot (N+1)$ as the number of groups per hidden unit $N_g$, can vary from 1 to $N+1$. The gradient of loss $E$ with respect to $z_{k,c}$ is,

$$g_{\text{Amolf}}(k,c) = \frac{\partial E}{\partial z_{k,c}} = -\frac{2}{N_v}\sum_{p=1}^{N_v}\sum_{i=1}^{M}(t_p(i) - y_p(i))\frac{\partial y_p(i)}{\partial z_{k,c}} \tag{5}$$

$$\frac{\partial y_p(i)}{\partial z_{k,c}} = w_{oh}(i,k)f'(n_p(k))\frac{\partial n_p(k,c)}{\partial z_{k,c}} \tag{6}$$

$$n_p(k,c) = \sum_{a\in c} x_p(i_k(a))[w(k,i_k(a)) + z_{k,c}g(k,i_k(a))] \quad \text{and} \quad n_p(k) = \sum_{c=1}^{Ng} n_p(k,c) \tag{7}$$

$$\frac{\partial n_p(k,c)}{\partial z_{k,c}} = \sum_{a\in c} x_p(i_k(a))g(k,i_k(a))] \tag{8}$$

Using Gauss-Newton updates, the elements of the 4-D Hessian $H_{Amolf}$ are computed as,

$$h_{Amolf}(k,c_1,j,c_2) = \frac{2}{N_v}\sum_{i=1}^{M} w_{oh}(i,k)w_{oh}(i,j)\sum_{p=1}^{N_v} f'(n_p(k))f'(n_p(j))\frac{\partial n_p(k,c_1)}{\partial z_{k,c1}}\frac{\partial n_p(j,c_2)}{\partial z_{j,c2}} \tag{9}$$

The 4 dimensional Hessian $\mathbf{H_{Amolf}}$ matrix is converted to 2 dimensional matrix as,

$$h_{Amolf}\left((k-1)N_g + c_1, (j-1)N_g + c_2\right) = h_{Amolf}(k,c_1,j,c_2) \tag{10}$$

The 2 dimensional gradient matrix $\mathbf{g_{Amolf}}$ is converted into column vector as,

$$g_{Amolf}\left((k-1)N_g + c\right) = g_{Amolf}(k,c) \tag{11}$$

The Gauss-Newton update guarantees that $\mathbf{H_{Amolf}}$ is non-negative definite. Given the negative gradient vector,

$$\mathbf{g_{Amolf}} = \left[ -\frac{\partial E}{\partial z_{1,c_1}}, -\frac{\partial E}{\partial z_{1,c_2}}, \ldots, -\frac{\partial E}{\partial z_{N_h,c_{N_g}}} \right]^T \tag{12}$$

$\mathbf{H_{Amolf}}$ is minimize $E$ with respect to vector $\mathbf{z}$ using Newton's method. The learning factors $\mathbf{z}$ can be computed as,

$$\mathbf{z} = \mathbf{H_{Amolf}^{-1}} \cdot \mathbf{g_{Amolf}} \tag{13}$$

The input weights matrix $\mathbf{W}$ are updated as follows,

$$w(k, i_k(a)) = w(k, i_k(a)) + z_{k,c} g(k, i_k(a)) \quad \text{where} \quad a \in c \tag{14}$$

Note that $\mathbf{H_{Amolf}}$ is quite compact that streamline computations compared to conventional Hessian computations.

### 3.4 Adapt-MOLF Initialization

The initial stage of Adapt-MOLF involves selecting the number of groups per hidden unit by exhaustively experimenting across the range of possibilities, from 1 to $N + 1$. The hyperparameter configuration yielding the lowest error is chosen as the starting point. This initialization technique significantly enhances algorithmic performance, outperforming random initialization approaches notably. In our experiments, periodic application, once every 50 iterations during training, further refines the algorithm's performance.

Evaluating errors for all potential group counts incurs considerable computational expense. To remove this, Adapt-MOLF uses interpolation between the Hessian and negative gradients with the Gauss-Newton Hessian and input weight negative gradients, respectively, as demonstrated by equation 13. This interpolation allows for the derivation of $\mathbf{H_{Amolf}}$ and $\mathbf{g_{Amolf}}$ from the Gauss-Newton Hessian and negative gradients $\mathbf{H}$ and $\mathbf{g}$ for input weights in the Adapt-MOLF algorithm. Expanding equation 5 and 9, we get

$$g_{\text{Amolf}}(k, c) = \sum_{a \in c} g(k, i_k(a))^2 \tag{15}$$

$$h_{\text{Amolf}}(k, c_1, j, c_2) = \sum_{a \in c_1} \sum_{b \in c_2} h(k, i_k(a), j, i_j(b)) g(k, i_k(a)) g(j, i_j(a)) \tag{16}$$

From equation 15 and 16, we can interpolate the Hessian and negative gradients of an Adapt-MOLF for any number of groups from Newton's Hessian and negative gradients of input weights. This helps to avoid recalculating Hessian and negative gradients for all the possible numbers of groups in determining the initial point of the algorithm.

### 3.5 Mathematical Treatment

*Lemma 1*: Assume $E(w)$ is a quadratic function of the input weight column vector $\mathbf{w}$ which is divided into k partitions $\mathbf{w}_k$ such that $\mathbf{w} = [\mathbf{w}_1^T, \mathbf{w}_2^T, \ldots, \mathbf{w}_k^T]^T$ and $g_k = \frac{\partial E}{\partial w_k}$. If a training algorithm minimizes $E$ with respect to the k dimensional vector $\mathbf{z}$ producing an error $E_k = E(\mathbf{w}_1 + z_1\mathbf{g}_1, \mathbf{w}_2 + z_2\mathbf{g}_2, \ldots, \mathbf{w}_k + z_k\mathbf{g}_k)$ and k can only increase by splitting one of the existing partitions, then $E_k + 1 \le E_k$

*Proof*: The error $E(\mathbf{w})$ after updating the input weights can be modeled as,

$$E(\mathbf{w} + \mathbf{e}) = E_0 - \mathbf{e}^T \mathbf{g} + \frac{1}{2} \mathbf{e}^T \mathbf{H} \mathbf{e} \tag{17}$$

where, $E_o$ is the error before updating the input weights, $\mathbf{g}$ is $\mathbf{g_k}$ for k=1, $\mathbf{H}$ is the Hessian, and $\mathbf{e}$ is the input weight change vector. If $\mathbf{e}$ is found optimally using Newton's method, then $\mathbf{e} = \mathbf{H}^{-1} \cdot \mathbf{g}$

The input weight change vector for k groups is

$$\mathbf{e}_k = [z_1\mathbf{g}_1^T, z_2\mathbf{g}_2^T, \ldots, z_k\mathbf{g}_k^T]^T \tag{18}$$

Given, $\mathbf{z} = argmin_z(E(\mathbf{w} + \mathbf{e}_k))$, increase k by one so that.

$$\mathbf{e}_{k+1} = [z_1\mathbf{g}_1^T, z_2\mathbf{g}_2^T, \ldots, z_{ka}\mathbf{g}_{ka}^T, z_{kb}\mathbf{g}_{kb}^T]^T \tag{19}$$

If $z_{ka} = z_{kb} = z_k$, then $\mathbf{e}_k = \mathbf{e}_{k+1}$ and $E_{k+1} = E_k$. However, since all the k+1 elements in $\mathbf{z}$ can change, we get $E_{k+1} \leq E_k$. *Lemma 1* presents a clear justification for increasing the number of second order groups or unknowns.

*Lemma 2*: If $E(\mathbf{w})$ is quadratic in each iteration, and if $E - E_{molf}$ and $E - E_{Amolf}$ denote the error decrease due to the Newton steps of OWO-MOLF and Adapt-MOLF respectively, then $E - E_{molf} \leq E - E_{Amolf}$

*Proof*: The $k$ groups of unknowns for Adapt-MOLF can be formed by splitting the $N_h$ groups of OWO-MOLF. The lemma follows from *Lemma 1*.

*Lemma 3*: OWO-Newton is a limiting case of the Adapt-MOLF algorithm as the k groups of Adapt-MOLF are split until $k = N_h \cdot (N + 1)$

*Proof*: We have

$$\mathbf{e_{Newton}} = \begin{bmatrix} z_1 \cdot \mathbf{g}_1 \\ z_2 \cdot \mathbf{g}_2 \\ \vdots \\ z_{N_h(N_g+1)} \cdot \mathbf{g}_{N_h(N_g+1)} \end{bmatrix} \tag{59}$$

In each iteration's Newton steps, the resulting errors adhere to the relationship $E_{\text{Newton}} \leq E_{\text{Amolf}}$. This assertion aligns with *Lemma 1*. *Lemmas 2* and *3* elucidate the Adapt-MOLF algorithm's interpolation between OWO-MOLF and OWO-Newton. The pseudo-code for the Adapt-MOLF procedure is delineated in Algorithm 1. Within the training process of the Adapt-MOLF algorithm, the grouping of weights associated with a hidden unit is determined by the second derivative of the loss concerning the weight, referred to as "curvature" in this context. Notably, this second derivative corresponds to the diagonal elements within the Hessian matrix. The Adapt-MOLF algorithm presents an intriguing approach with the potential to streamline model complexity.

### 3.6 Computational Burden

The computational burden is used to measure the time complexity for each algorithm. It indicates number of multipliers that a particular algorithm needs to process per iteration using inputs, hidden units and outputs. We calculate computational burden for Adapt-MOLF, OWO-MOLF, SCG and LM. The proposed Adapt-MOLF algorithm involves calculating Hessian. However, compared to Newton's method or LM, the size of the Hessian is much smaller. Updating input weights using Newton's method or LM, requires a Hessian with $N_w$ rows and columns, whereas the Hessian used in the proposed Adapt-MOLF has only $N_h$ rows and columns. The total number of weights in the network is denoted as $N_w = M \cdot N_u + (N + 1) \cdot N_h$ and $N_u = N + N_h + 1$. The number of multiplies required to solve for output weights using the OLS is given by $M_{ols} = N_u(N_u + 1)[M + \frac{1}{6}N_u(2N_u + 1) + \frac{3}{2}]$. Therefore, the total number of multiplications per training iteration for LM, SCG, OWO-MOLF and Adapt-MOLF algorithm is given as :

$$M_{lm} = [MN_u + 2N_h(N+1) + M(N + 6N_h + 4) + MN_u(N_u + 3N_h(N+1)) + 4N_h^4(N+1)^2] + N_w^3 + N_w^2 \tag{20}$$

$$M_{scg} = [MN_u + M(N + 6N_h + 4) + MN_u(N_u + 3N_h(N + 1)) + 4N_h^4(N + 1)^2] + N_w^3 + N_w^2 \tag{21}$$

$$M_{owo-molf} = M_{ols} + N_v N_h [2M + N + 2 + \frac{M(N_h + 1)}{2}] \tag{22}$$

$$M_{adapt-molf} = M_{owo-molf} + N_v [N_h(N + 4) - M(N + 6N - N_h + 4)] + (N_h)^3 \tag{23}$$

Note that $M_{adapt-molf}$ consists of $M_{owo-molf}$ plus the required multiplies for calculating adaptive learning factors. The computational cost of Adapt-MOLF algorithm varies between computational cost of OWO-MOLF and OWO-Newton.

---

**Algorithm 1** Adapt-MOLF algorithm

---

1: Read the training data. Initialize $\mathbf{W}$, $\mathbf{W_{oi}}$, $\mathbf{W_{oh}}$, $N_{it}$ , $N_g \leftarrow N_h$, $i_t \leftarrow 0$,
2: **while** $i_t < N_{it}$ **do**
3:     Compute $\mathbf{G}$ and do **Adapt-MOLF steps** :
4:     a: Compute $\mathbf{L}$ (the curvature of error w.r.t input weights) from equation (3). $size(\mathbf{L}) = (N_h, N + 1)$
5:     b: $\mathbf{I} \leftarrow argsort(\mathbf{L}, axis = 1)$. Input weight indices sorted in the descending order of curvature. $size(\mathbf{I}) = (N_h, N + 1)$
6:     c: Divide the sorted indices of weights connected to each hidden unit into $N_g$ groups of equal size.
7:     d: Compute Adapt-MOLF learning factors $\mathbf{z_{Amolf}}$ from equation (13). $size(\mathbf{z_{Amolf}}) = (N_h \cdot N_g, 1)$
8:     e: Create learning factor column vector $\mathbf{z}$ by applying same learning factors for all the weights in a group. $size(\mathbf{z}) = (N_h \cdot (N + 1), 1)$
9:     Update input weights: $\mathbf{W} \leftarrow \mathbf{W} + \text{diag}(\mathbf{z}) \cdot \mathbf{G}$
10:     **OWO step** : Solve equation (2) to obtain $\mathbf{W_o}$ and compute $EPM_{i_t}$ from equation (4)
11:     $N_g \leftarrow N_g + (\text{sign}(EPM_{i_t} - EPM_{i_t - 1}))$ ; $i_t \leftarrow i_t + 1$
12: **end while**

---

## 4 Experimental Results

The Adapt-MOLF algorithm is empirically evaluated with OWO-MOLF, SCG and LM using *Twod* (32), *Single2* (32), *Oh7* (32) and *Mattern*(32) datasets. The data sets used for the simulations are listed in Table 1.

Table 1: Description of Data Sets used in Experiments

| Data Set Name | No. of Inputs | No. of Outputs | No. of Patterns |
|---|---|---|---|
| Twod | 8 | 7 | 1768 |
| Single2 | 16 | 3 | 10000 |
| Oh7 | 20 | 3 | 15000 |
| Mattrn | 4 | 4 | 2000 |

The data sets are normalized to zero mean before training. The number of hidden units to be used in the MLP is determined by network pruning using the method of (33). By this process the complexity of each of the data sets is analyzed and an appropriate number of hidden units is found. Training is done on the entire data set 10 times with 10 different initial networks. The average Mean Squared Error (MSE) from this 10-fold training is shown in the plots below. The average training error and the number of multiplies is calculated for every iteration in a particular dataset using the different training algorithms. These results are then plotted to provide a graphical representation of the efficiency and quality of the different training algorithms.

For the *Twod* data file MLP is trained with 27 hidden units. In Figure 2, the average mean square error (MSE) from 10-fold training is plotted versus the number of iterations for each algorithm. In Figure 3, the average training MSE from 10-fold training is plotted versus the required number of multiplies (shown on a

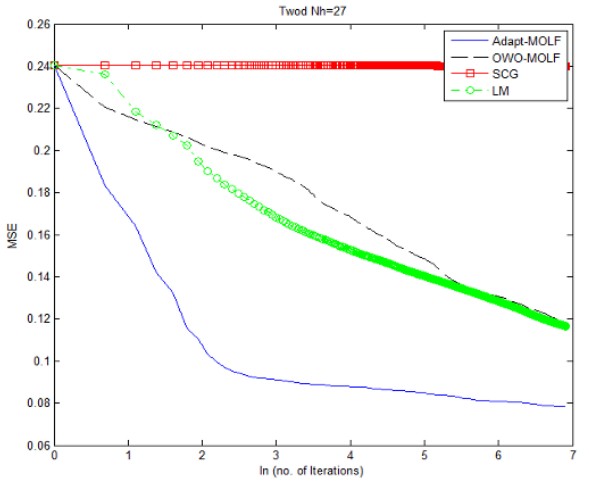

Figure 2: Twod.tra data set: average error vs. number of iterations

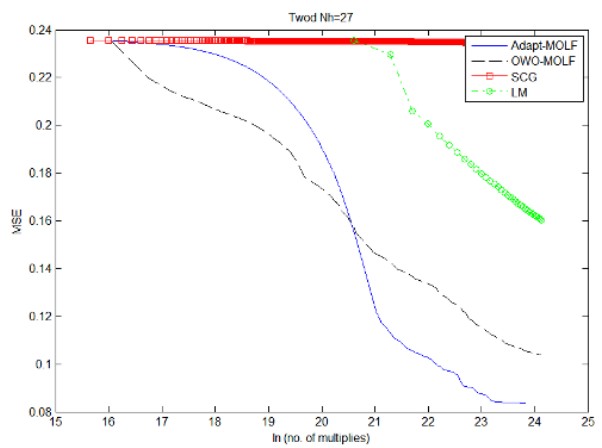

Figure 3: Twod.tra data set: average error vs. number of multiplies

log scale). From these plots, Adapt-MOLF is performing far better than the other three algorithms both in terms of number of iterations and number of multiples.

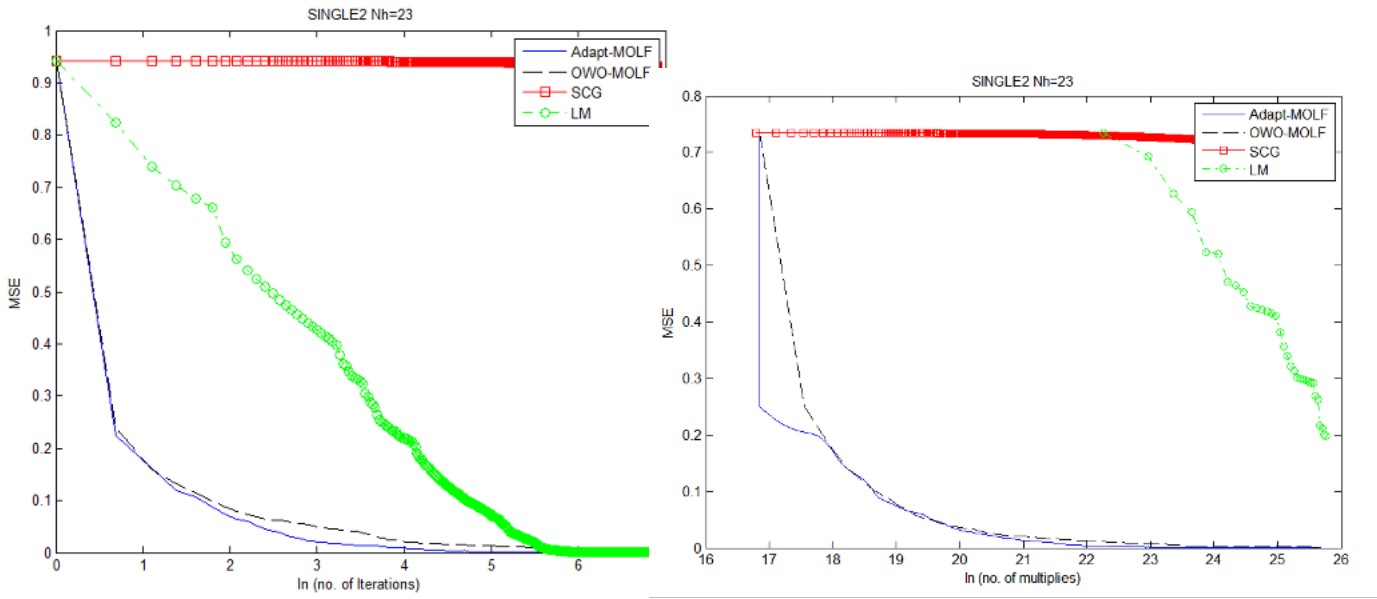

Figure 4: Single2.tra data set: average error vs. number of iterations

Figure 5: Single2.tra data set: average error vs. number of multiplies.

For the *Single2* dataset, the MLP is trained with 23 hidden units. In Figure 4, the average mean square error (MSE) from 10-fold training is plotted versus the number of iterations for each algorithm (shown on a loge scale). In Figure 5, the average training MSE from 10-fold training is plotted versus the required number of multiplies (shown on a loge scale). For this dataset the performance of Adapt-MOLF is very close to that of OWO-MOLF. This shows that the number of groups per hidden unit $N_g$ is 1 in most of the iterations.

For the *Oh7* dataset, the MLP is trained with 23 hidden units. In Figure 6, the average mean square error (MSE) for training from 10-fold training is plotted versus the number of iterations for each algorithm (shown on a loge scale). In Figure 7, the average training MSE from 10-fold training is plotted versus the required

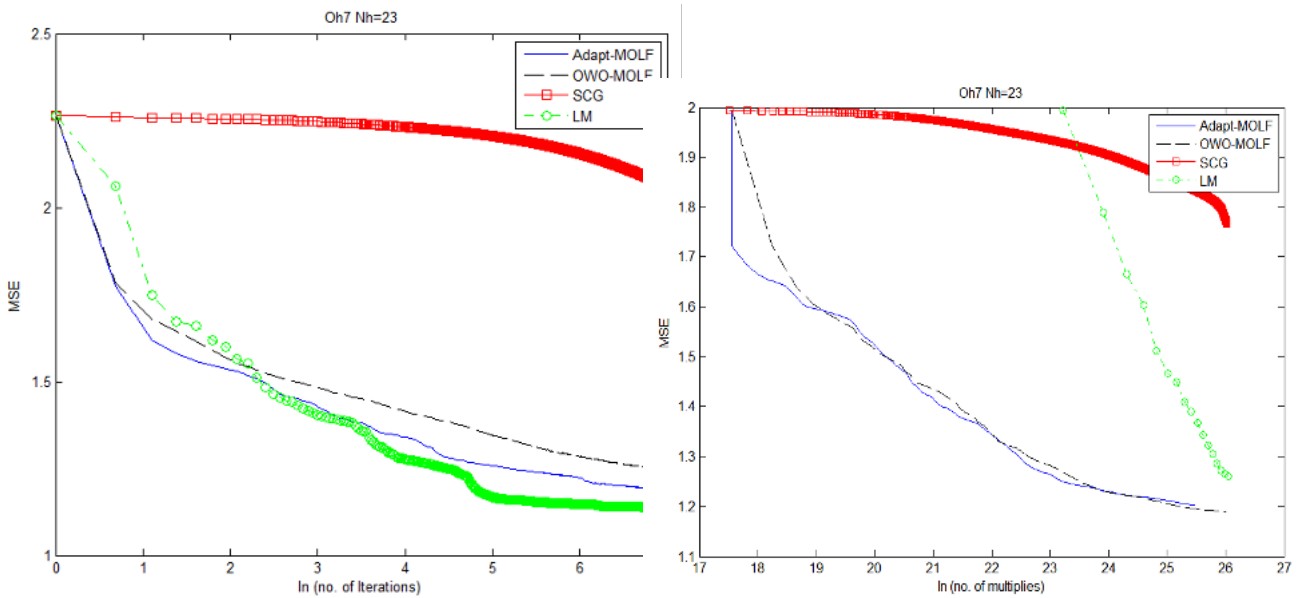

Figure 6: Oh7.tra data set: average error vs. number of iterations

Figure 7: Oh7.tra data set: average error vs. number of multiplies.

number of multiplies (shown on a loge scale). In this data set the proposed algorithm is performing better than OWO-MOLF in terms of iterations. In terms of multiples the performance of Adapt-MOLF is very close to that of OWO-MOLF.

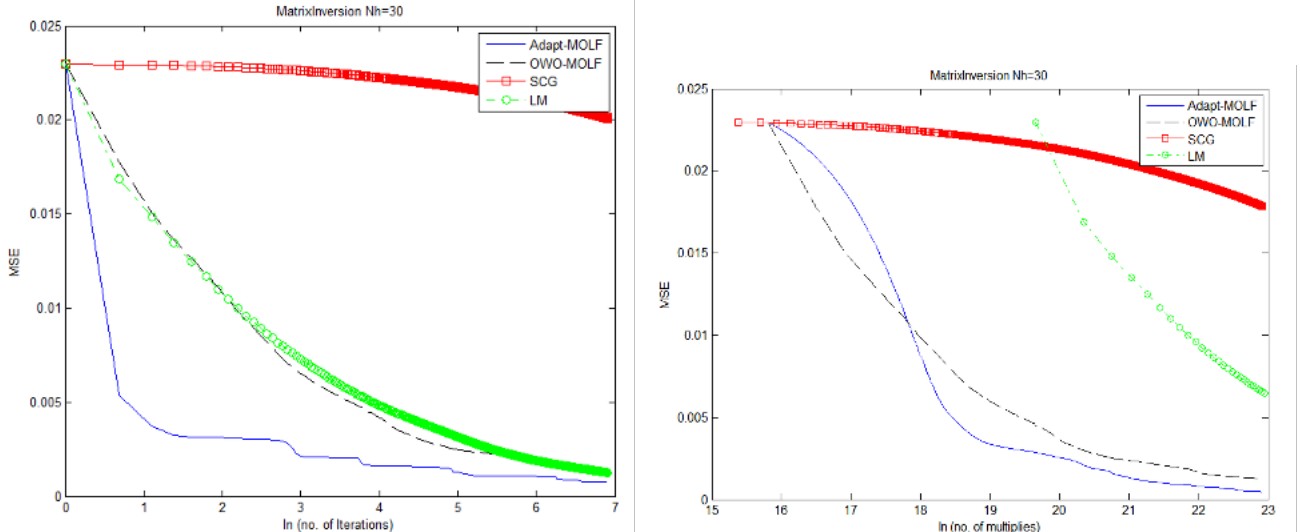

Figure 8: Matrix inversion data set: average error vs. number of iterations

Figure 9: Matrix inversion data set: average error vs. number of multiplies

For the matrix inversion data file [8], the MLP is trained with 30 hidden units. In Figure8, the average mean square error (MSE) from 10-fold training is plotted versus the number of iterations for each algorithm (shown on a loge scale). In Figure 9, the average training MSE from 10-fold training is plotted versus the required number of multiplies (shown on a loge scale). For this dataset the proposed algorithm is dominating the other three algorithms.

### 4.1 K-fold Validation and Testing

The k-fold validation procedure is used to obtain the average training and testing errors. In k-fold validation, given a data set, it is randomly split into k non-overlapping parts of equal size, of which (k-2) parts are used for training, one part for validation and the remaining one part for testing. In this technique the training is stopped when we get a satisfactory validation error, and the resulting network will be tested on the testing data to obtain the test error. This procedure is repeated k times to obtain average training and testing errors. For the simulations the k value is chosen as 10. The Table 2 depicts Adapt-MOLF's better performance compared to other methods in terms of iteration count and convergence operations. SCG and LM exhibit comparatively higher errors, suggesting potential limitations in handling these specific datasets. OWO-MOLF show low training but higher testing errors, indicate potential overfitting. While OWO-MOLF maintains stable performance across datasets, it dosen't performance best in specific scenarios. Adapt-MOLF achieved comparable performance with OWO-MOLF using nearly the same number of operations and comparable performance with LM with less than 50% of operations required for LM.

Table 2: 10-fold training ($E_{trn}$) and testing ($E_{tst}$) errors. Best-performing results are in bold

| DataSet | Adapt-MOLF | OWO-MOLF | SCG | LM |
|---------|------------|----------|-----|-----|
| Twod | **0.0888 / 0.1172** | 0.1554 / 0.1731 | 1.0985 / 1.0945 | 0.2038 / 0.2205 |
| Single2 | **0.0042 / 0.0175** | 0.0151 / 0.1689 | 3.5719 / 3.6418 | 0.0083 / 0.0178 |
| Mattrn | **0.0011 / 0.0013** | 0.0027 / 0.0032 | 4.2400 / 4.3359 | 0.0022 / 0.0027 |
| Oh7 | **1.2507** / 1.4738 | 1.3205 / 1.4875 | 4.1500 / 4.1991 | 1.1602 / **1.4373** |

Table 2 presents 10-fold training and testing errors showcasing the Adaptive MOLF algorithm's consistent and competitive performance. Across the datasets, Adaptive MOLF is a top-performing algorithm or closely competes with other methods, demonstrating its robustness and effectiveness. Notably, in the *Twod* dataset, Adaptive MOLF achieves significantly lower errors in both training and testing compared to OWO-MOLF, SCG, and LM, emphasizing its superiority. In the *Single2* dataset, Adaptive MOLF displays exceptional performance by yielding the lowest testing error among all algorithms, showcasing its remarkable generalization capabilities. Moreover, in the *Mattrn* dataset, Adaptive MOLF consistently produces the lowest errors in both training and testing phases. Despite a higher training error in the *Oh7* dataset, Adaptive MOLF excels by achieving the best testing error, underlining its robust generalization prowess. Overall, the results strongly suggests that the Adaptive MOLF algorithm consistently competes favorably or outperforms other algorithms, particularly excelling in testing errors, signifying its potential for better generalization across diverse datasets.

## 5 Conclusion and Future Work

The Adaptive Multiple Optimal Learning Factors (Adapt-MOLF) algorithm presented in this research successfully mitigates the scalability challenges inherent in second-order training algorithms. This is achieved through an adaptive mechanism that modulates the computation of learning factors using Newton's method, thereby enhancing the efficiency of error reduction per multiplication operation. Empirical evaluations indicate that the Adapt-MOLF algorithm exhibits superior performance compared to the OWO-MOLF algorithm, both in terms of error reduction per iteration and often in error decrease per multiply. Furthermore, the algorithm demonstrates a unique capacity to interpolate between the OWO-MOLF and OWO-Newton methodologies. The scope of this study is intentionally focused, applying the concept of Adaptive Multiple Optimal Learning Factors exclusively to input weights in a Multilayer Perceptron with a single hidden layer. This concentration allows for a detailed exploration and elucidation of the algorithm's derivation and operational specifics. Looking forward, subsequent research will expand the application of the Adapt-MOLF algorithm to more complex deep neural network architectures. This will facilitate a comprehensive comparison of its performance against established first-order neural network optimization techniques. This forthcoming analysis is anticipated to provide further insights into the efficacy and applicability of the Adapt-MOLF algorithm within the broader context of neural network optimization.

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
