# OpenReview forum: "Adaptive Multiple Optimal Learning Factors for Neural Network Training"
_TMLR — Withdrawn by Authors_

### Review · Reviewer_YiPv · 2024-01-14

**Summary Of Contributions:**

The authors introduce a second-order optimization method that uses second order information to limit the number of parameters that are optimized using curvature-based heuristics to group parameters for optimization and Gauss-Newton updates. The authors introduce an "error per multiply" (EPM) figure of merit and use that to decide the number of groups for optimization "N_g".

**Audience:**

Yes

**Claims And Evidence:**

No

**Requested Changes:**

Please explain:

* What does it mean for a convergence rate to “align with the number of unknowns.”

* Regarding focusing on a two-layer MLP, you say, “This concentration allows for a detailed exploration and elucidation of the algorithm’s derivation and operational specifics”. I am not sure why it is so hard to extend your analysis for deeper models.

* What does “No. of patterns” identify in Table 1?

Please modify:

* A lot of equations in Section 3.3 can be moved to the appendix.

* The quality of Figure 1 is really poor. Currently, it’s just a screenshot of a figure.

* It is not clear what the background is used for. What do we consider all methods in Section 2.3? We need some propert motivation.
The introduction in Section 3 can come handy in introducing that.

**Strengths And Weaknesses:**

Strengths:

* The authors introduce Algorithm 1 that is able to outperform the baselines on several tasks (e.g. in Figures 2 and 3).

* The idea of using curvature to decide which parameters should be optimized is sound.

Weaknesses:

* The main text is extremely hard to follow due to the dense notation. The intuition and motivations behind the mathematical manipulations are hidden across the text, and not emphasized at all. The extra complication comes from the fact that there are many typos, e.g. there is a parenthesis missing in Equation (22).

* The significance and the semantics of the datasets in the experiments is not explained at all. The reader is expected to be familiar with these datasets. I am not sure the claim “The Adaptive Multiple Optimal Learning Factors (Adapt-MOLF) algorithm presented in this research successfully mitigates the scalability challenges inherent in second-order training algorithms” is validated in the paper.

* Figures 4-7: AdamMOLF is very close to OWO-MOLF, and also LM is clearly winning in Figure 6. Are these results statistically significant? Error bars are missing.

* Any analysis on how EPM and N_g change with time is missing, which is a pity, since these variables are introduced by the algorithm.

---

### Review · Reviewer_DsUw · 2024-01-18

**Summary Of Contributions:**

The paper proposes an Adapt-MOLF, a second-order optimization algorithm that clusters the input weights of multiplayer perceptron (MLP) and only computes the relevant statistics within each cluster. This cluster is dynamically adjusted throughout training, which the authors claim is advantageous compared to existing structured second-order techniques. Empirically, the authors investigate the effectiveness of Adapt-MOLF in simplistic settings (e.g., MLP with a single hidden layer), and show improved performance against baseline techniques such as scaled conjugate gradient (SCG).

**Audience:**

Yes

**Broader Impact Concerns:**

I do not believe the work requires a Broader Impact Statgement.

**Claims And Evidence:**

No

**Requested Changes:**

- In the introduction, the authors motivate the work by describing (1) the challenges of second-order approximation for neural networks and (2) the limitations of existing optimization techniques such as AdaFactor [1] and LAMB [2]. However, I find that some claims, such as the applicability of AdaFactor not well understood, are unreasonable. This can be said for any neural network optimizers, such as stochastic gradient descent. The last sentence briefly describes the algorithm proposed in the paper, but it does not connect to the arguments the authors made in the previous paragraphs. I recommend revising the introduction to make the contribution more clear.
- Figure 1, describing the fully connected MLP, does not provide much information. I suggest removing or adding more content that describes the set-up of the paper studies.
- \citep and \citet are not correctly used. I understand this was done to reduce the space in the paper, but it makes it extremely difficult to follow the paper. For example, does (3) refer to equation 3 or reference 3?
- The descriptions of the proposed algorithm in Section 3 can be improved.
- I recommend that the authors provide more details of their experimental setup in the Appendix and compare the algorithm with other optimizers. It would be helpful to include a plot, where the x-axis is the wall-clock time, to see the relative computational comparisons.

**Strengths And Weaknesses:**

- The writing in the paper can be significantly improved. There are various formatting (e.g., omitting equations or references) and notation issues, which makes it extremely challenging to follow the paper. Apart from these problems, it is also difficult to follow the overall flow of the paper (which I will further elaborate on in the section below).
- Many claims in the paper are not well justified (which I detail in the section below).
- The authors study a limited setting: multilayer perceptron (MLP) with sigmoid activation, a single hidden layer, and MSE loss. The setting also assumes the residual connection for the MLP architecture (which the authors term “fully connected MLP”). However, this is not clearly communicated in the abstract or introduction and is not justified. For example, the author refers to the above specific architecture and setting as MLP, which is technically incorrect.
- The authors do not describe the limitations of their approach, such as scalability and limited algorithm applicability.
- The empirical study has several limitations. Firstly, the authors do not compare their algorithm to other second-order methods mentioned before, such as Shampoo. Given that the network size is small (e.g., 30 hidden units), it would be helpful to include the results using the exact Hessian (without block structure). Especially in this limited setting the author explores, I would expect a more detailed analysis and comparison to other existing approaches. Secondly, a detailed description of the experimental setup is missing from the Appendix (e.g., hyperparameter used), and the code is not provided, which makes the algorithm difficult to reproduce.
- At this stage, it is challenging to understand the applicability of Adapt-MOLF and its relative performance to other optimizers.

---

### Review · Reviewer_UZ37 · 2024-01-18

**Summary Of Contributions:**

The paper proposed Adapt-MOLF, a second-order optimization algorithm that dynamically calculates the number of weight groups for efficiently achieving the Hessian matrix.  The proposed method is an extension of OWO-MOLF algorithm where instead of keeping the group size fixed, it adaptively calculate the group size based on the curvature of the error function. With the proposed method, the paper claims that it achieves faster convergence rate than OWO-MOLF algorithm and less computational cost than LM. The paper also empirically showing that the proposed algorithm can achieve faster or comparable convergence rate with a MLP on different datasets.

**Audience:**

Yes

**Claims And Evidence:**

No

**Requested Changes:**

1. The plots should be more clear and at least with the same size.

2. For Figure 2 and 3, why LM is worse than the proposed method and OWO-MOLF? Can we have some intuition of that?

3. For figure 4 and 5. The OWO-MOLF and Adapt-MOLF actually have similar convergence rate for iteration numbers and based on the paper, OWO-MOLF should be cheaper in computational costs. Then why the convergence rate is much faster for Adapt-MOLF in figure 5?

4. LM seems to achieve smaller MSE in figures against iteration numbers but why it has higher for the multipliers? Can we have a full plot? Like training longer.

5. Can we have more experiments on different widths of the MLP to see the trade-offs between the costs and convergence? In addition, can we have the training time for all these experiments?

6. Is there a chance that we can apply the proposed methods to other deep learning architectures?

**Strengths And Weaknesses:**

Strengths: The idea is simple and based on the experiments, the proposed method is effective.

Weakness:

1. The idea is not novel enough.
2. The paper is not well written. The plots are not well presented and lack intuition and analysis for the experiment results.
3. I am concerned about the practical effectiveness since the paper only shows that the proposed method can only work on a 2-layer MLP.

---

### Note · Authors · 2024-02-23

**Comment:**

We would be improving upon the paper based on the comments.

**Withdrawal Confirmation:**

I have read and agree with the venue's withdrawal policy on behalf of myself and my co-authors.